# Public–Private Partnership: Participants’ Experiences of the Web-Based Registration-and-Management System for Patients with Hypertension and Diabetes Mellitus

**DOI:** 10.3390/healthcare11091297

**Published:** 2023-05-01

**Authors:** Seonah Lee, Nanyoung Song, Suhyun Kim

**Affiliations:** 1College of Nursing, Chonnam National University, Gwangju 61469, Republic of Korea; salee@jnu.ac.kr; 2Hypertension & Diabetes Registration Center, Suwan Health Life Support Center, Gwangju 62247, Republic of Korea; indana1828@daum.net; 3Department of Nursing, Nambu University, Gwangju 62271, Republic of Korea

**Keywords:** public–private partnership, web-based registration and management system, hypertension and diabetes mellitus, public health

## Abstract

Hypertension and diabetes mellitus, which induce cardiovascular and cerebrovascular diseases, require government intervention. In South Korea, the web-based registration-and-management system for patients with hypertension and diabetes mellitus was operated as a pilot program. This study explored the experiences of the 71 participating medical centers in G city, South Korea, of using the web-based registration-and-management system for preventing and managing hypertension and diabetes mellitus. After the survey, 40 physicians participated in interviews, and the recorded interviews were analyzed and classified into three categories: participation motivation, participation experiences, and suggestions. The study participants participated in a national pilot project with different expectations. Similar to the survey results on participation experiences, the satisfaction from a patient-care perspective was high, but there was an excessive burden of administrative work from the perspective of primary-medical-center operations. In addition, the suggestions included strengthening systematic education, continuous maintenance, broad project application, and improvements to the system. The system needs to be unified and simplified, the project continuity needs to be secured, and the application of the system to other regions and medical centers needs to be considered to induce public–private cooperation to reduce the amount of administrative work, which is currently excessive.

## 1. Introduction

Cardiovascular and cerebrovascular diseases are major health concerns worldwide and are among the leading causes of death [1]. Hypertension and diabetes mellitus are two important risk factors for these diseases [2]. Therefore, preventing and managing hypertension and diabetes mellitus at a national level is crucial to reduce the burden of cardiovascular and cerebrovascular diseases on individuals and society [2,3]. These conditions are also significant contributors to healthcare costs [4]. However, daily prevention and management can help to reduce the incidence of cardiovascular and cerebrovascular diseases and related medical expenditures. Thus, national-level interventions are necessary [3,5].

To address this, the South Korean government implemented a comprehensive plan for the prevention and management of hypertension and diabetes mellitus, which included a pilot program for a web-based registration-and-management system [5,6]. 

This program aimed to establish a local-community-centered management system by promoting public–private partnerships (PPPs), led by the private sector [5,6]. The PPPs can leverage additional resources from the private sector and gather the influence of local community leaders to address local issues [5,6]. Successful pilot-project results require private medical centers’ cooperation and participation [7,8].

The Korean government started the hypertension-and-diabetes registration-management project in one city in 2007, expanded it to 19 regions in 2012, and to 25 public health centers in 11 cities and provinces in 2020 [8]. In this program, public institutions, such as public health centers identify patients with hypertension and diabetes mellitus and register them in the management system. These institutions also provide educational and statistical materials to private medical centers and take charge of public health education and public relations in local communities [8,9]. Furthermore, private medical centers provide the continual treatment and management of registered patients, including the identification and registration of such patients and providing community health centers with patient information [8,9]. In 2012, the participation rate of medical institutions was 19.4%, and that of pharmacies was 24.5%. In 2020, the participation rates of medical institutions and pharmacies increased to 44% and 62.0%, respectively [8,10]. The evaluation of the program’s effects and the development of future strategies require a comprehensive understanding of the experiences of private medical centers [10]. 

However, there have been no specific data regarding the participation experiences of a web-based registration-and-management system for preventing and managing hypertension and diabetes mellitus by public–private cooperating institutions. Only an efficacy assessment [11] and the participation experiences of residents [12] have been reported. Thus, this study explored and analyzed experiences of participation in a web-based registration-and-management system for preventing and managing hypertension and diabetes mellitus at private medical centers through surveys and detailed interviews, which were conducted to obtain more objective data. These data can provide primary information on policy implications regarding the effective operation and direction of participating institutions in the future. 

## 2. Materials and Methods

### 2.1. Study Design

This was a mixed-method study using self-administered surveys and interviews [13,14].

### 2.2. Participants

In a local community in G city in South Korea, 85 primary medical centers participated in a web-based registration-and-management system for patients with hypertension and diabetes mellitus. In this study, 71 medical centers participated in the survey, and 40 physicians participated in in-depth interviews. The selection criteria for participating institutions were persons in charge of institutions that expressed voluntary participation intention and persons in charge of institutions that participated in the hypertension-and-diabetes mellitus registration-and-management system for more than one year (Figure 1).

### 2.3. Data Collection

After obtaining the Institutional Review Board (IRB)’s approval, the data were collected for one month from 1 December 2020. Text messages were sent to interested institutions to confirm their participation in the study. Because this pilot project was a national public–private endeavor, the cellphone numbers of the persons in charge of participating institutions were known. After confirming participation by asking persons in charge of participating institutions, the participating medical centers were visited and provided with some introductory information and consent forms. 

The data collected were quantitative and qualitative. For the quantitative data, a self-administered questionnaire used in a pilot-project-efficacy report was revised [15]. A web link directing users to the revised 18-question self-administered questionnaire (Appendix A) was sent to participants, who confirmed their voluntary participation via text message. Each survey took about 10–15 min to complete each.

The contents of the questionnaire were as follows. Regarding the project status, participation intention and registration-system input were included. Regarding project efficacy, the number of patient visits, treatment sustaining rate, medication compliance, interest in public health education, utilization of educational materials, overall satisfaction, and project support were included. Furthermore, the general characteristics included the ages of study participants, types of medical specialty, duration of clinic operation, the average number of patients with hypertension per day, the average number of patients with diabetes mellitus per day, and the occupations of study participants. 

In-depth interviews were conducted to collect qualitative data, and introduction, development, and conclusion questions were included. A draft was created by analyzing the preexisting research, and questions were revised for readability and study objectives through consultation with two nursing professors, who had experience in qualitative studies, to prepare the questions for the interviews in this study. The reviewed questions were revised after they were shown to three employees of this project’s participating institution. To avoid disturbing the treatment time due to sudden visits, the doctors’ interview times were scheduled in advance, and the visits were made accordingly. The interviews were conducted once the treatments ended. The interviews took place in the doctors’ offices because the interviews needed to be held in quiet and comfortable rooms for study participants. Interview questions were used; each interview lasted about 60–80 min. During each interview, an investigator recorded notes. The investigator double-recorded the interview through a smartphone and recording device and obtained consent from each participant prior to recording. After confirming that no more new answers to the questions would be given, the interview was completed. The interview questions, including main and supporting questions, are summarized in Appendix A.

### 2.4. Data Analysis 

This study analyzed data obtained through self-administered questionnaires and interviews to comprehensively understand the participation experiences of primary medical centers of web-based hypertension-and-diabetes registration-and-management system. Using the MS Excel program, quantitative data, such as the participants’ demographical data and survey responses, were analyzed by calculating the frequency and percentage. 

We used NVivo 11 (QSR International) to manage all data (meeting reports, structured observations, interview transcripts, and data obtained from document analysis), including the qualitative data obtained after the interviews, and a thematic analytical approach was applied [16,17]. The coding framework was developed through consultations, including themes from the data and the topic-guide questions. The transcripts were read, the data were coded, and the generated charts of grouped themes were exported to MS Word to explore emerging patterns for further interpretation. For the subanalysis, charts were used to analyze the grouped themes and patterns with a view to defining the major categories better, and they became the basis for the organization and presentation of the study’s findings. 

### 2.5. Data Analysis Study Validity and Rigor

Following Lincoln and Guba [18], trustworthiness was secured when analyzing the data to validate the qualitative study and maintain rigor. First, the recorded data and interviews were compared to ensure consistency in securing the actual values. Second, the study results were sent to three physicians at primary medical centers that did not participate in this study, and the similarities between and results of the reported experiences were reviewed to ensure the applicability of the findings. When the large study was completed, representatives from the health facilities and district health-management teams that participated attended feedback meetings (n = 4), and the study findings were shared. The research team and the representatives discussed and evaluated the results’ applicability through these meetings. Third, the data-analysis process and procedures were complied with thoroughly, and the study results were reviewed and assessed by five qualitative-study specialists to assess the study’s process and ensure its consistency. The research team sent the specialists the study methods, procedures, and results and requested a consultation and a confirmation of the study’s validity and rigor. When analyzing the study’s results, prejudice and bias in the participants’ experiences were excluded as much as possible to ensure neutrality. For neutrality, the interview was conducted by a researcher with a good practical understanding of the pilot program. Interview analysis was conducted by researchers who were well-versed in qualitative research but had no personal prejudice or prior understanding of the operation and evaluation of this pilot program.

### 2.6. Ethical Considerations

This study was approved by H University’s Institutional Review Board (IRB no. 1040198-200824-HR-095-02). Furthermore, the study’s objective and participation methods were explained to participants volunteering for the study. Upon provision of the study description, written consent was obtained through the study-participation-consent form. All interviews were recorded, and it was explained to the participants that interviews could be terminated if they wanted to stop for any reason. Each participant who filled in the questionnaire or completed the interview received a gift worth about KRW 5000.

## 3. Results

The total number of medical centers that participated in this study was 71. Regarding the physicians of the medical centers that participated in this study, six were aged 30–39, nineteen physicians were aged 40–49, seventeen were aged 50–59, and sixteen ere aged 60–69. Most (31) were internists. There were 35 medical centers that had been established for more than 10 years, of which 33 had used the hypertension-and-diabetes-mellitus registration-and-management system for more than five years (Table 1). 

After implementing the hypertension-and-diabetes-mellitus registration-and-management system, 69% (49) of the participants reported increased regular visits, and 79% (56) reported improved medication compliance. In addition, 81% of the participants (57) reported improved blood pressure and blood-glucose control, while 81% (58) reported increased interest in diseases. Moreover, because of the hypertension-and-diabetes-mellitus registration-and-management system, 84% of the participants (62) reported improved mutual relationships with patients. The education materials on the hypertension-and-diabetes-mellitus registration-and-management system regarding drug administration were utilized by 85% (60) of the participants. Overall, 86% of the participants (61) were satisfied with the hypertension-and-diabetes-mellitus registration-and-management system (Figure 2).

As a result of the in-depth interviews on the participants’ experiences of the hypertension-and-diabetes-mellitus registration-and-management system, the major categories confirmed in this study were (1) participation motivation, (2) participation experiences, and (3) suggestions (Table 2).

### 3.1. Participation Motivation

#### Participation due to Expectations as Part of a National Pilot Project

The subcategory of participation motivation was “Participation because of expectations as part of a national pilot project.” This subcategory comprised four themes: patient-centered project, support from the Korean Medical Association (KMA), the expectation of an increase in treatments, and the prevention of the loss of patients to other hospitals. The study participants decided to participate for their patients and, because the KMA announced their support for this project, their participation was driven by trust. Moreover, because the treatment fees and drug expenses were covered, the patients were expected to visit the medical centers more often. Furthermore, the study participants partook in this project out of fear of losing their patients to other hospitals if they did not.

*“If my medical center receives support for treatments and medications by participating in the hypertension-and-diabetes-mellitus registration-and-management system, it is good”*.

*“A monthly payment of KRW 1,500 might seem little, but an older patient needs to continue treatment, so I participated in this project”*.


*“As a physician who operates a medical center, because the government supports medical costs once per month, patients come more frequently instead of coming to the medical center once per two months, and more patients, as well as treatment cases, were expected by participating in this project”.*



*“Although I was not sure what it was, it was a project initiated by the government and supported by the KMA, so I participated in it”.*



*“I participated in this project because I was afraid that my patients would go to different medical centers to receive the treatment fee and drug expenses if I did not participate”.*


### 3.2. Participants’ Experiences

The participants’ experiences of the web-based hypertension-and-diabetes-mellitus registration-and-management system comprised two subcategories: the patient-care perspective and the primary-medical-center-operation perspective. 

#### 3.2.1. Patient-Care Perspective 

This subcategory was composed of six themes: the increase in the number of patients’ visits to medical centers, drug-administration-compliance improvement, the maintenance of treatment continuity, improvements in blood-pressure and blood-glucose control, improved mutual relationships with patients, and increases in patients’ interest in diseases. The study participants noticed that the number of their patients’ visits to the medical centers increased, and their medication compliance increased as a result. Furthermore, as they had conversations with their physicians more frequently, the patients’ lifestyles improved, and their treatment continuity was maintained, improving their blood-pressure and blood-glucose control. Above all, because of their participation in this project, the treatment fee and drug expenses were supported, so the patients were less financially burdened, allowing them to respond more favorably to recommendations or suggestions. The study participants took part in education sessions held at centers with the hypertension-and-diabetes-mellitus registration-and-management system and showed a greater interest in managing their diseases on their own. 

Increase in the number of patients’ visits to medical centers:

*“Actually, because of the hypertension-and-diabetes-mellitus registration-and-management system, the number of patients increased. Especially, more patients aged 65 years or older have been coming to see me. Supporting the medical cost for one visit per month seems very effective”*.

*“After participating in the hypertension-and-diabetes-mellitus registration-and-management system, the number of patients visiting my medical center increased a lot. When I asked them to come to see me once per month, they willingly came to see me and did not have any burden because the government paid KRW 1,500, which covered the cost”*.

Drug-administration-compliance improvement:

*“Because of the increased number of regular visits to the medical center, medication compliance improved as well”*.

Maintenance of treatment continuity:

*“Treatment is maintained because patients continually come to the medical center once per month, although it seems obvious. Anyway, for me, the benefit is in treating patients continually”*.

Improvements in blood-pressure and blood-glucose control:

*“As the number of visits to the medical center increased, and medication compliance improved, patients tended to have more interest in their own blood pressure and blood glucose, thereby improving their control over them”*.

Improved mutual relationships with patients:

*“Because the medical center does not require patients to pay for their visits, they listen better. They come to the medical center regularly to receive support from the hypertension-and-diabetes-mellitus registration-and-management system, so our relationship with patients improves”*.

*“The best thing about participating in the hypertension-and-diabetes-mellitus registration-and-management system is that patients do not complain about having a regular visit once per month. In the past, it often caused issues with my patients. Regular visits and treatment are important for chronic diseases, but patients misunderstood that they were for money”*.

Increases in patients’ interest in diseases:

*“Yesterday, one patient told me he attended an education session held by the hypertension-and-diabetes-mellitus registration-and-management center. I certainly felt that patients were interested in their diseases and tried to manage them because of the hypertension-and-diabetes-mellitus registration-and-management system”*.

*“When an older patient, who received education from the hypertension-and-diabetes-mellitus registration-and-management center, asked about his hemoglobin A1C level, I was amazed. It means they are more interested in their own diseases because of the hypertension-and-diabetes-mellitus registration-and-management system”*.

#### 3.2.2. Primary-Medical-Center-Operation Perspective

The primary-medical-center-operation-perspective subcategory has two themes: the burden of excessive administrative work and satisfaction with project participation. The study participants, who were medical professionals taking care of their patients and administrators of their medical centers, described their experiences of participating in this project. Because the study participants operated primary medical centers, their medical centers were often small, so it was common for their employees to leave their jobs. Thus, conducting training on the hypertension-and-diabetes-mellitus registration-and-management system whenever the person in charge left was challenging. Some of the study participants input details into the system and managed them themselves after work; others hired employees who were in charge of encoding this information. Most of the study participants were satisfied with their participation in the hypertension-and-diabetes-mellitus registration-and-management system because their patients benefited, the number of treatment visits increased, and their relationships with their patients improved. 

The burden of excessive administrative work:

*“At my medical center, nurses input information into the hypertension and diabetes-mellitus registration-and-management system. In addition, nurses recommend that patients participate in the hypertension-and-diabetes mellitus registration-and-management system and give them information. The workload of nursing staff increased.”*.

*“If a nurse does not enter accurate information into the hypertension-and-diabetes mellitus registration and management system, a problem occurs in the treatment. Thus, while seeing patients, I directly enter information into the hypertension and diabetes mellitus registration and management system and explain the project”*.

*“I, a physician, enter information myself into the hypertension and diabetes mellitus registration and management system. Because this system is not linked to the treatment chart system, it requires much work. Also, nursing assistants quit their jobs very frequently, so it is very tedious and burdensome to teach new nursing assistants how to manage the hypertension-and-diabetes-mellitus registration-and-management system. Therefore, I do it myself even if it means I have to go to work on my day off”*.

Satisfaction with project participation

*“**Actually, this project helps me to operate the medical center and maintain my practice. Since the opening of the medical center, the medical cost has been covered for patients who were happy about it. Also, I was very satisfied as well”*.

*“**Because of the hypertension**-and-diabetes-mellitus registration-and-management system, the treatment cost and the complex test fees have been supported, so I think South Korea is a good country. In addition, older people keep thanking me, although I did not pay for them, so I feel good”*.

*“**In my opinion, the best thing about the hypertension**-and-diabetes-mellitus registration-and=management center was the education on hemoglobin A1C. It takes much time to explain this test alone. Because patient education is performed regularly by the hypertension**-and-diabetes-mellitus registration-and-management system, I am very happy about it”*.

*“Of course, I would recommend this project to my colleagues or perform it in other regions because patients receive great benefits”*.

### 3.3. Suggestions

The suggestions fell into the following subcategories: the necessity of strengthening the systematic education project, the broad application of the project, and the expansion of the age groups of the participants.

#### 3.3.1. The Necessity of Strengthening the Systematic Education Project

The subcategory of the necessity of strengthening the systematic education project was composed of eight themes (Table 3). In clinical practice, the study participants noticed significant differences between the knowledge and education levels of the patients with hypertension and diabetes mellitus. Despite performing patient education through the hypertension-and-diabetes-mellitus registration-and-management system, systematic education was requested for treating the patients. 

The importance of measuring blood pressure and blood glucose at home:

“Many patients respond too sensitively regarding their blood-pressure or blood-glucose levels. Informing them on how to measure them at home would be helpful”.

*“Especially, older patients with diabetes mellitus want everything to be cared for at the medical center. However, it is not possible. Please inform them of the importance and accurate method of self-measuring blood glucose”*.

The necessity of regular tests:

*“There are still patients who consider testing very negatively, although not many of them nowadays. Nobody wants to argue with them. So, I hope that the hypertension-and-diabetes-mellitus registration-and-management center will inform them of which tests are needed and why”*.

The importance of continual treatment:

*“**Some patients tend not to come to the medical center anymore or stop taking medications alone if they feel their health is improving. It is sad that they come back to the medical center if their health deteriorates. Please emphasize this point. It is essential not to stop treatment”*.

The necessity of vaccination for patients with chronic diseases:

“Sometimes, patients agree to receive vaccination against pneumonia for free but do not agree if they need to pay for it. They believe we recommend it for money. So, if the hypertension-and-diabetes-mellitus registration-and-management center educates them, such patients will receive vaccinations more willingly”.


*“*
*Because immunity is low in patients with chronic diseases, vaccination is essential, but patients are often ignorant of it. Thus, the importance of vaccination must be educated*
*”.*


Education on complications of chronic diseases:

*“Education on complications of patients with hypertension and diabetes mellitus is necessary if they are controlled”*.

*“**For those who do not understand complications well, specific symptoms and risks must be educated repeatedly”*.

Diet for hypertension and diabetes mellitus:

*“Especially older people want to know what to eat and what to avoid eating specifically. Such education will give practical help”*.

*“Vague education will make patients eat whatever they want. So, please use a picture of a table setting or other food pictures for education”*.

Diet for older people:

*“**Diets might be different for older people. Because of their tooth condition or blood-glucose levels, there is a limitation. So, there must be specific education on diet”*.

Easy home exercises:

“It would be great to receive training from a personal trainer in a gym, but educational materials for home exercises would be helpful”.

*“Everybody knows doing exercises is a good thing, but older people usually do not do exercises. If they are taught what to do step by step, in detail, they can easily follow them at home”*.

#### 3.3.2. Continual Maintenance and Broad Application of the Project

The subcategory of the continual maintenance and broad application of the project comprised three themes: securing continuous possibilities, expanding the regions of the project’s implementation, and expanding the age groups of the participants. The study participants felt nervous because this was a pilot project. Since the patients received benefits as a result of the project, they would not have easily accepted the sudden termination of the project. Moreover, because this pilot project certainly provided benefits to the patients, its expansion to other regions and broadening the age groups of the participants are desirable.

*“The biggest challenge was the continuity of the hypertension and diabetes mellitus registration and management system. The web-based hypertension and diabetes mellitus registration and management system was perceived as an unstable pilot project”*.

*“Nationwide, there are not many regions where this project is performed. I do not understand why this good project has been expanded to other towns in the city or other regions in the country. It provides patients with education, complication tests, and free tests. Also, it even educates and manages the medical centers”*.

*“A policy is needed to increase the enrollment rate of not only people aged 65 years or older but also those aged between 30 and 64 years”*.

#### 3.3.3. System Improvement

The subcategory of system improvement comprised two themes: the improvement of the testing system and the simplification of the input system. The study participants sometimes wanted to perform one test only, but for the patients in the project, only a set of grouped tests could be performed, so they found it inconvenient. Thus, they wanted to improve the testing system. In addition, since the participants experienced inconveniences related to the system input, the simplification of the input system and the development of a program that unifies the preexisting medical-insurance-request system or disease-management system are recommended.

Improvement of the testing system:

*“When seeing patients, sometimes testing for a cholesterol level is all I need. However, the testing system of the hypertension-and-diabetes-mellitus registration-and-management system only allows the performance of a set of grouped tests. If needed, it would be better if it were possible to perform only a cholesterol test”*.

Simplification of the input system:


*“The biggest problem is that there is a lot of administrative work. The medical insurance request system, disease management system, and hypertension and diabetes mellitus registration and management system are operated separately, so we must enter information into each system. Thus, there is a great burden of entering information into the hypertension-and-diabetes-mellitus registration-and-management system. I feel sorry for nurses as well. Simplification of administrative work and unification of systems are needed”.*


## 4. Discussion

This study was performed to explore experiences of the web-based system for the prevention and management of hypertension and diabetes mellitus at PPP institutions by analyzing the integrated results of a survey and interviews. To successfully implement the web-based registration-and-management system, partnerships with the medical centers in local communities are essential, so it is important to understand the experiences of the medical centers participating in this project. 

First, regarding their motivation for participating in the hypertension-and-diabetes-mellitus registration-and-management system, most of the participants answered that the hypertension-and-diabetes-mellitus registration-and-management system was patient-oriented. They thought that this project benefited patients significantly [15]. In clinical practice, supporting patients with the treatment fees and drug costs encouraged regular patient visits, thereby increasing the number of treatment cases. Because the hypertension-and-diabetes-mellitus registration-and-management system was a pilot project performed by the government, not all medical centers participated, and only some medical centers participated voluntarily. Based on the participation statuses of the medical centers, the patients who visited the medical centers that did not participate in this project did not receive any benefits. Currently, some patients ask their medical centers first whether they participate in the hypertension-and-diabetes-mellitus registration-and-management system to establish whether they can receive support. Therefore, because of the fear that discontinuing their participation in this project may result in the loss of patients, some medical centers continued to participate. Thus, as the study participants suggested, the recruitment of more medical centers for participation is difficult unless continuity is secured. The medical centers currently participating in this project have concerns regarding the project’s continuity. They are nervous about its possible unexpected termination in the future because a sudden termination of the project would cause confusion among their patients and, furthermore, the burden would be borne by the medical centers themselves. Therefore, the government must secure the project’s continuity to encourage the participation of the medical centers. 

As shown by the results of the frequency analysis and interviews regarding the participants’ experiences of the hypertension-and-diabetes-mellitus registration-and-management system, an increase in the number of patients’ visits to their medical centers, drug-administration-compliance improvements, the maintenance of treatment continuity, the improvements in blood-pressure and blood-glucose control, improved mutual relationships with patients, and increases in patients’ interest in diseases were commonly observed [11,15,19]. The Elderly Healthcare Voucher Scheme has been performed in Hong Kong since 2009, increasing private primary-healthcare-service utilization [10]. However, the demand for primary public healthcare has not decreased. Basic questions remain to be answered, such as the scheme’s potential role in improving universal healthcare coverage and the financial maintenance associated with the program’s structure. In addition, the exploration of better or additional mechanisms to effectively approach more complex healthcare priorities, including chronic disease management, is necessary.

Although a multidisciplinary approach may improve patient care, because of the increased burden of administrative work involved in the operation of the medical centers, the satisfaction levels of the participants were double-sided. This project provided various benefits for the patients, such as the support for the treatment fees and drug costs but not for the participating medical centers. Therefore, encouraging long-term participation in the project and PPP may be difficult [11,19]. 

The study participants had difficulties in entering information into the system because of frequent workforce changes, and additional system-management work beyond treatment activities may prevent individuals from participating in this project. In clinical practice, even if those in charge of projects are replaced, through education on input systems using e-learning programs, the burden on medical centers can be reduced [20]. Moreover, setting an additional budget for human resources would benefit participating institutions when this project is performed in the future. 

Lastly, regarding the problems and the suggestions noted by the primary medical centers after experiencing the hypertension-and-diabetes-mellitus registration-and-management system, despite the systemic education delivered by the system, the medical centers recommended strengthening this education even further. Since the patients receiving the education were old, the importance of repeated education was highlighted [21]. Although the contents, which the study participants wanted to strengthen, were already included in the educational materials of the hypertension-and-diabetes-mellitus registration-and-management system, the level of education was still insufficient because the number of people involved in education in clinical practice was smaller than the number of people who received support for their treatment fees and drug costs. For managing patients with hypertension and diabetes mellitus in particular, investing in education is the beginning of the prevention of larger complications. In reality, in South Korea, few facilities specialize in education compared to the number of patients; only 31 community health centers are registered to conduct educational training sessions on blood pressure and diabetes mellitus [20]. As a strategy to more effectively manage and prevent chronic diseases, academia–practice partnerships are also needed to supplement and broadly apply education programs. Firm academia–practice partnerships may be crucial for improving evidence-based decision making and applying evidence-based programs and policies [22]. Specifically, regarding exercise-related education, a specialized exercise program may be provided by linking with the Elderly Voucher Customized Sports Service [23]. The preexisting Elderly Voucher Customized Sports Service has reduced medical expenditure and improved disadvantaged older people’s well-being through health promotion, emotional recovery, improvements in quality of life through the formation of better relationships, and reduced visits to medical centers [23]. The medical centers requested that the age groups of the participants be expanded by considering the ages of all the patients with hypertension and diabetes mellitus. It is important to prevent and manage chronic diseases, so registration and management seem necessary even for young patients. It is worthwhile to consider expanding the current voucher project, which has been successful, as a strategy for chronic-disease management. 

Although this study was conducted during the COVID-19 pandemic, the private institutions participating in it did not notice the impact of COVID-19 on their use of the web-based registration-and-management system for the prevention and management of hypertension and diabetes mellitus. In fact, previous studies also showed that the blood-pressure- and blood-sugar-control rates of subjects who received registration management for hypertension and diabetes in 2020 continued to increase compared to the previous year [24]. This observation means that the web-based registration-and-management system can be effectively operated through cooperation with local governments even during the pandemic period of an infectious disease, such as COVID-19, which was made possible in this case through effective non-face-to-face education for subjects registered on the web, including mobile messaging apps or SNS, YouTube channels, online training-completion sites, videoconferencing (Zoom), or quizzes on blood-pressure and blood-sugar management using SNS platforms. It is also possible to actively promote access to education and business using various methods, such as the transmission of video links and educational videos [25]. In addition, among registered patients, older patients who complain of difficulties in using the Internet can increase their participation by engaging in person-specific education on Internet use prior to the project’s implementation with a 1:1 reservation system at an education center [25]. Therefore, it is necessary to develop non-face-to-face educational materials and strategies to increase patients’ engagement with the educational topics suggested by the public–private partnerships in this study.

## 5. Conclusions

This was a mixed-method study, in which experiences of the web-based registration-and-management system for the prevention and management of hypertension and diabetes mellitus were analyzed at PPP institutions. This study was meaningful because the members of the PPP medical centers who participated in the project were invited to accurately assess the hypertension-and-diabetes-mellitus registration-and-management system. The study’s results were valuable as they offer basic data on participants’ experiences of hypertension-and-diabetes-mellitus management at private medical centers, which could be used to formulate policies aiming at the effective operation and direction of participating institutions in the future. The private medical centers were satisfied with this pilot project in terms of patient care, but they felt burdened by the additional work of entering information into the system without an incentive. To prevent and manage chronic diseases and draw continuous PPPs, the unification and simplification of the input system must be ensured soon, and a detailed plan for expanding the project to other regions and medical centers needs to be discussed.

## Figures and Tables

**Figure 1 healthcare-11-01297-f001:**
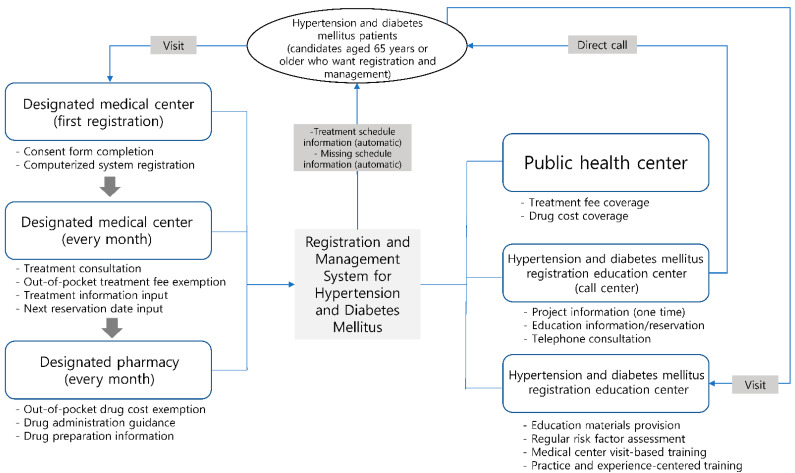
Web-based hypertension-and-diabetes-mellitus registration-and-management-system model.

**Figure 2 healthcare-11-01297-f002:**
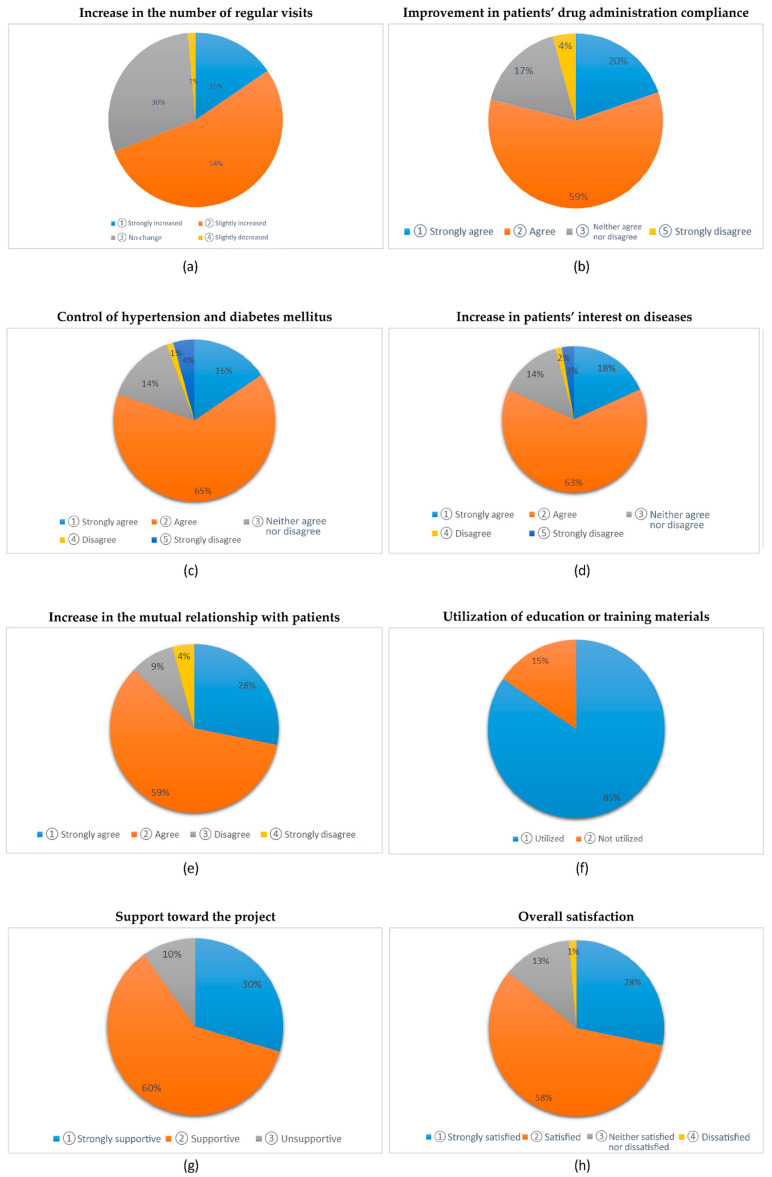
Frequency analysis of the institutions’ perceptions and attitudes after participating in hypertension-and-diabetes-mellitus registration-and-management system: (**a**) the increase in the number of regular visits; (**b**) improvement in patients’ drug-administration compliance; (**c**) control of hypertension and diabetes mellitus; (**d**) increase in patients’ interest in diseases; (**e**) increase in the mutual relationship with patients; (**f**) utilization of education or training materials; (**g**) support for the project; (**h**) overall satisfaction.

**Table 1 healthcare-11-01297-t001:** General characteristics of participating institutions (n = 71).

Category		Frequency (%)
Age groups of the medical-center directors	30 s	6 (8.4)
	40 s	25 (35.2)
	50 s	22 (31.0)
	≥60 s	18 (25.4)
Specialization	Internal medicine	37 (52.1)
	Family medicine	18 (25.4)
	Surgery	3 (4.2)
	Others	13 (18.3)
Duration of medical-center operation	<5 years	21 (29.6)
	5–9 years	8 (11.3)
	≥10 years	42 (59.1)
Participation period in the hypertension-and-diabetes-mellitus project	0.5–1 year	6 (8.4)
	1 year ≤ x > 3 years	12 (16.9)
	3 years ≤ x > 5 years	11 (15.5)
	≥5 years	42 (59.2)
The average number of patients with hypertension per day	≤9 patients	29 (40.8)
	10–19 patients	23 (32.4)
	20–29 patients	12 (16.9)
	≥30 patients	7 (9.9)
The average number of patients with diabetes mellitus per day	≤9 patients	34 (47.9)
	10–19 patients	24 (33.8)
	20–29 patients	7 (9.8)
	≥30 patients	6 (8.5)
The average number of patients with hypertension and diabetes mellitus per day	≤9 patients	30 (42.2)
	10–19 patients	23 (32.4)
	20–29 patients	8 (11.3)
	≥30 patients	10 (14.1)
The occupations of the respondents	Physicians	43 (60.6)
	Nurses/Nursing assistants	21 (29.6)
	Others	7 (9.8)

**Table 2 healthcare-11-01297-t002:** Physicians’ experiences of web-based hypertension-and-diabetes registration-and-management system at primary medical centers.

Category	Subcategory	Themes
Participation motivation	Participation due to expectations, as part of a national pilot project	Patient-centered project Support from the Korean Medical Association (KMA) The expectation of an increase in the number of treatments Prevention of loss of patients to other hospitals
Participation experiences	Patient-care perspective	Increase in the number of patients’ visits to medical centers Drug-administration-compliance improvement Maintenance of treatment continuity Improvement in blood pressure and blood-glucose control Improved mutual relationships with patients Increase in patients’ interest in diseases
	Primary-medical-center-operation perspective	The burden of excessive administrative work Satisfaction with project participation
Suggestions	The necessity of strengthening systematic education	Importance of measuring blood pressure and blood glucose at home The necessity of regular tests Importance of continual treatment The necessity of vaccination for patients with chronic diseases Education on complications of chronic diseases Diet for hypertension and diabetes mellitus Diet for older people Easy home exercises
	Continued maintenance and broad application of the project	Securing continuous possibilities Expanding regions of project implementation Expanding age groups of participants
	System improvement	Improvement of the test system Simplification of the input system

**Table 3 healthcare-11-01297-t003:** Training list, with an emphasis on improving the systematic education project.

Recommended Training Topics
The importance of measuring blood pressure and blood glucose at home The necessity of regular tests The importance of continual treatment The necessity of vaccination for patients with chronic diseases Education on the complications of chronic diseases Diet for hypertension and diabetes mellitus Diet for older people Easy home exercises

## Data Availability

The data presented in this study are not publicly available due to privacy reasons.

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
