# Peer review of "Public–Private Partnership: Participants’ Experiences of the Web-Based Registration-and-Management System for Patients with Hypertension and Diabetes Mellitus"

_healthcare, 2023, doi:10.3390/healthcare11091297_

Round 1

Reviewer 1 Report

The manuscript by Lee et al presents the results of the survey of the 71 medical centers from G city, South Korea, members of which have participated in the government-supported pilot effort aiming to improve medical care of patients with hypertension and diabetes via newly introduced web-based registration and management system. Well conducted and presented, the study reports the overall improvement in the patient care satisfaction and (based on the responses of participating physicians) provides valuable suggestions as to how further increase the efficiency of the system, reduce administrative burden for physicians, strengthen systematic education etc. In the reviewer’s opinion, the importance of this study goes beyond the analysis of this pilot effort, as its conclusions might be expanded to further medical care in other medical systems and countries. In line with the last notion, and to present the message of the study to a broader audience, the reviewer suggests to add following minor modifications:

1)          It will be informative to add a short paragraph to the ‘Introduction’ section, summarizing (in the chronological order) the key steps in the nation-wide diabetes and hypertension treatment programs implemented by the South Korean government;

2)          Include the complete version of the questionnaire (as a supplementary data file);

3)          Figure 2: add headlines to the separate panels to improve readability (i.e. add the caption ‘the increase in the number of regular visits’ on top of the Figure 2a, etc.);

4)          Results: given the importance of the patient education further highlighted by this study, the key points from the section 3.4.1 ‘The necessity of strengthening the systematic education project’ should be presented as the separate figure/table (to increase their visibility);

5)          Discussion: given that the study has been conducted during the COVID pandemic, to add a sentence or more discussing whether and how the web-based registration and management system might facilitate care for this particularly vulnerable group (>65 y, hypertension, diabetes).

Reviewer 2 Report

Overall, this is an interesting research that has potentials to get published, but there are some concerns regarding the theoretical background and conclusion, which lead my suggestion to be "Major revision".

1、 Introduction

(1) What is the novelty of this study over and above what has already been addressed by past studies? The authors should elaborate the contributions in a much clearer way.

(2) The introduction is very Korean-centric. Please note that the journal has an international readership of behavioral researchers. The audience is mainly interested in psychological phenomena, not specific national situations. Therefore, please put the phenomenon at the core of your manuscript, not Korea.

2、 Materials and Methods

The methods are not appropriate and not sufficiently detailed to be reproducible. This section may contain sufficient detail so that, when read in conjunction with cited references, all procedures can be repeated. In particular, 2.3 2.4.2 2.5. The interview questionnaire should be presented as an attachment.

3、 Results

(1)The authors abuse the use of paragraphs and subsections. When a reasonable number of them are used, they allow the information to be structured. However, if they are abused, they end up hindering the understanding of the paper.

(2)The title of each paragraph should clearly help readers better understand the content, In particular,3.1.1 3.1.2.

I hope that all these issues are understood in a positive sense and that being aware of the mistakes made in this work will allow the authors to be aware of these problems in order to avoid them in their future papers.

Round 2

Reviewer 2 Report

It is ok to publish.